# CCN1-Mediated Signaling in Placental Villous Tissues after SARS-CoV-2 Infection in Term Pregnant Women: Implications for Dysregulated Angiogenesis

Yuyang Ma [†], Liyan Duan [†], Beatrix Reisch, Rainer Kimmig, Antonella Iannaccone [‡] and Alexandra Gellhaus *,[‡]

Department of Gynecology and Obstetrics, University Hospital Essen, 45147 Essen, Germany; yuyang.ma@stud.uni-due.de (Y.M.); duanliyan1988@126.com (L.D.); beatrix.reisch@uk-essen.de (B.R.); rainer.kimmig@uk-essen.de (R.K.); antonella.iannaccone@uk-essen.de (A.I.)
* Correspondence: alexandra.gellhaus@uk-essen.de
[†] These authors contributed equally to this work.
[‡] These authors also contributed equally to this work.

**Abstract:** The global spread of SARS-CoV-2 has increased infections among pregnant women. This study aimed to explore placental pathology alterations and angiogenic factor levels in term pregnant women after SARS-CoV-2 infection in a retrospective single-center study. Additionally, we investigated the role and underlying mechanism of the vascular inflammation-promoting, cysteine-rich protein 61 (CYR61/CCN1) in this context. All analyses were performed in term pregnant women infected with or without SARS-CoV-2. The sFlt-1, PlGF, and sEng serum levels were quantified using ELISA. Placental protein expressions were examined by immunoblot and immunostaining. Additionally, the effect of CCN1 protein on SGHPL-5 trophoblast cells was examined. We found that SARS-CoV-2 activated the inflammatory response in pregnant women, leading to pronounced vascular alterations in placental villous tissues. Elevated serum anti-angiogenic factors (sFlt-1, sEng) upon SARS-CoV-2 infection may directly contribute to these pathological changes. Upregulated CCN1 and pNF-κB in placental villous tissues of infected patients are identified as crucial factors in placental alterations. As a conclusion, CCN1 was significantly elevated in the placentas of term pregnant women infected with SARS-CoV-2. By activating a cascade of inflammatory responses, CCN1 induced the production of the anti-angiogenic factors sFlt-1 and sEng, which may lead to abnormal placental vascular architecture.

**Keywords:** CCN1; NF-κB; sFlt-1; sEng; pregnancy; SARS-CoV-2; placenta; trophoblast

## 1. Introduction

Some viruses, such as Zika virus and cytomegalovirus, can infect fetuses through the placental barrier and cause fetal malformations or even stillbirth [1–3]. Coronavirus disease 2019 (COVID-19), declared a pandemic on 11 March 2020 by the World Health Organization (WHO), showed as a causative agent severe acute respiratory syndrome coronavirus-2 (SARS-CoV-2), an enveloped, positive, single-stranded RNA virus [4]. At the beginning of the COVID-19 epidemic, most of the literature indicated that SARS-CoV-2 did not cause substantial increases in maternal and neonatal mortality [5,6]. However, evidence has emerged indicating that SARS-CoV-2-positive pregnant women are at an increased risk of preeclampsia and preterm birth, while fetuses may be at an increased risk of intrauterine distress, growth restriction, and serious neonatal complications [7,8]. Villar et al. identified that, among 416 newborns from infected women, 54 (13%) tested were positive and cesarean delivery but not breastfeeding was associated with an increased risk of neonatal positivity potentially by the transmission of infection after birth [7]. Additionally, SARS-CoV-2 has been detected in the placenta, amniotic fluid, umbilical cord, and fetal tissues of

SARS-CoV-2-positive patients [9]. McMahon et al. confirmed the presence of SARS-CoV-2 in fetal *mouse* neurons, glial cells, and choroid plexus cells from infected individuals, indicating that prenatal SARS-CoV-2 infection cases may have significant implications for the neurodevelopment and function of the offspring [10]. Histopathological data from placental tissues (fibrin deposition and inflammatory infiltrate in the intervillous space) assume a severe inflammatory response, resulting in the impairment of the fetal–maternal barrier. From an ultrastructural point of view, the virus-targeted placental cells appear to be trophoblasts and fibroblasts. Transmission electron microscopy data showed virus particles within the cytosol of placental cells. Furthermore, morphological data suggest that various factors (direct cytopathic, ischemic injuries and an inflammatory response) cooperate to compromise the physiological functions of the placenta, including gas exchange, metabolic transfer, hormone secretion, and fetal protection [11]. Therefore, it has been assumed that complications in fetuses or newborns may be attributed to aberrant placental functionality resulting from maternal infection with SARS-CoV-2.

Cysteine-rich protein 61 (CYR61/CCN1) belongs to the CCN family of extracellular matrix proteins involved in intercellular signaling especially in angiogenesis [12]. CCN1 can also promote inflammatory responses by binding to integrin CD11b to recruit monocytes, macrophages, and neutrophils [13–15]. CCN1 expression increased after the infection of astrocytes with certain viruses, such as Zika virus and coxsackievirus B3 [16,17]. Moreover, Kase and Okano demonstrated that CCN1 caused various dysfunctions of the central nervous system by enhancing the inflammatory response caused by SARS-CoV-2 in infected patients [18]. In addition, Forsyth et al. demonstrated that the SARS-CoV-2 spike protein subunit 1 induced high levels of NF-κB activation, resulting in pro-inflammatory cytokine production in *human* bronchial epithelial cells [19]. NF-κB is the downstream signaling factor of CCN1 promoting inflammation [20,21]. Therefore, it is reasonable that CCN1 and NF-κB may be also involved in SARS-CoV-2-induced histological alterations in the placenta.

Both soluble fms-like tyrosine kinase 1 (sFlt-1) and soluble endoglin (sEng) are anti-angiogenic factors that can be secreted by the placenta [22]. It is well known that those two factors are increased in preeclampsia and secreted into the maternal circulation [23,24]. Under normal conditions, vascular endothelial growth factor (VEGF) and placental growth factor (PlGF) bind to vascular endothelial growth factor receptor-1 (VEGFR-1/Flt-1) and vascular endothelial growth factor receptor-2 (VEGFR-2) to maintain the vasodilation and functional stability of placental vascular endothelium, while sFlt-1 can competitively bind to VEGF and PlGF to disrupt vascular homeostasis [22]. An elevated maternal sFlt-1/PlGF ratio is used as a clinical marker of angiogenic imbalance in preeclampsia [25]. Therefore, it is an indicator to assess placental function. In normal pregnancy, transforming growth factor beta (TGFβ) activates endothelial nitric oxide synthase (eNOS) by binding to transforming growth factor beta receptor (TGFβR) to promote vasodilation during pregnancy. Excessive sEng can block this pathway by binding to TGFβR1/2, resulting in sustained vasoconstriction [22,26].

Hosier et al. indicated that the placenta and umbilical cord of a 22$^{nd}$-week pregnant women with COVID-19 tested positive for SARS-CoV-2 ribonucleic acid, while fetal heart and lung tissue tested negative after intrauterine fetal death [27]. Some SARS-CoV-2-positive placentas presented chorionic inflammation, increased subchorionic fibrin deposition, intervillous thrombosis, and thickening of the vascular wall [28–30], which belong to histological manifestations of maternal vascular malperfusion (MVM) [31,32]. These changes may be due to the activation of the maternal systemic immune response caused by SARS-CoV-2, resulting in a cytokine storm and finally leading to MVM on the one hand and local inflammation of the placenta, on the other hand, directly caused by SARS-CoV-2 [33]. However, there are no studies so far reporting evidence of protein levels of placental inflammation markers in SARS-CoV-2-infected pregnant women. In this study, we aimed to investigate alterations in placental pathology and angiogenic factor levels in term pregnant women following SARS-CoV-2 infection, along with elucidating the underlying mechanisms. Therefore, we analyzed the expression of inflammatory factors CCN1

and pNF-κB expression in the placenta of term pregnant women infected with SARS-CoV-2 compared to uninfected controls. In addition, the expression levels of the anti-angiogenic factors sFlt-1 and sEng in serum and placentas were analyzed, and the correlation between CCN1 and the anti-angiogenic factors sFlt-1 and sEng in the trophoblast cell line SGHPL-5 was investigated.

## 2. Materials and Methods

Detailed descriptions of the methods are shown in the Supplementary Materials.

### 2.1. Study Population

Pregnant women were diagnosed with SARS-CoV-2 infection using nasopharyngeal RT-PCR analysis conducted at our hospital's Institute of Virology. In line with the WHO's classification of COVID-19 severity, pregnant participants were categorized as asymptomatic, mildly symptomatic, or severely symptomatic [34]. Additionally, all newborns enrolled in our study yielded negative results in nasopharyngeal swab tests for SARS-CoV-2 presence. The placental chorion villous tissues of term pregnancies after delivery were obtained from the Department of Gynecology and Obstetrics, University Hospital Essen, Essen, Germany. Written, informed consent for the women's participation in the study was obtained and the study was approved by the local ethics committee of the University of Duisburg-Essen (12-5212-BO, 21-10462-BO).

The placenta analysis involved 47 term pregnant women: 23 controls without known SARS-CoV-2 infection during pregnancy and 24 infected with SARS-CoV-2 between 36 and 41 weeks of gestation. Patient characteristics are summarized in Table 1. For all patients infected with SARS-CoV-2, the time from infection to delivery was less than 14 days. In the SARS-CoV-2-positive group, asymptomatic infection accounted for 62.5%, mild infection accounted for 37.5%, and no severe infection occurred. In addition, in the control group, which contained gestational-age-matched cases, all patients were not infected by SARS-CoV-2. There were two cases of gestational diabetes mellitus (GDM) and one case each of hypertension, oligohydramnios, fetal growth restriction (FGR), and portal hypertension. In the SARS-CoV-2 group, there were four cases of GDM and one case each of diabetes mellitus, hepatitis B, and anemia.

**Table 1.** Comparative analysis of clinical characteristics in maternal SARS-CoV-2 infection: impact on term pregnancy placental cohort.

| Variable | Control (n = 23) | SARS-CoV-2 (n = 24) | *p*-Value |
|---|---|---|---|
| Maternal age at delivery, years, median (IQR) | 34.00 (30.00–37.00) | 32.50 (29.25–36.50) | 0.4257 |
| Gestational age at delivery, weeks, mean (min and max) | 38 + 6 (36 + 2–41 + 0) | 39 + 0 (36 + 5–41 + 1) | 0.8336 |
| Pregnancy BMI before birth, median (IQR) | 31.00 (27.00–35.00) | 28.50 (26.25–32.75) | 0.3159 |
| Cesarean section, no. (%) | 14 (60.87) | 11 (45.83) | - |
| Leukocyte, $10^3/mm^3$, mean ± SD | 10.98 ± 2.96 | 9.07 ± 3.44 | 0.0363 |
| CRP, mg/dL, mean ± SD | 0.39 ± 0.42 | 2.54 ± 3.61 | 0.0197 |
| Asymptomatic infection, no. (%) | - | 15 (62.50) | - |
| Mild infection, no. (%) | - | 9 (37.50) | - |
| Birth weight, g, mean ± SD | 3387 ± 392.70 | 3570 ± 529.90 | 0.3853 |
| 5-min Apgar score, mean (min and max) | 9.87 (9–10) | 9.58 (8–10) | 0.0686 |

Abbreviation: BMI, body mass index; CRP, C-reactive protein; IQR, interquartile range; SD, standard deviation. Note: Gestational age is expressed as a combination of weeks and days. For example, 38 + 6 means 38 weeks plus 6 days.

In a separate patient cohort, divided into control and SARS-CoV-2 groups of 14 patients each, the serum levels of angiogenic proteins sFlt-1, PlGF, and sEng were measured. All patients delivered at our institution and were infected with SARS-CoV-2 in the third trimester of pregnancy. Since we did not obtain blood from all patients where we analyzed the placentas, we created a second cohort of patients for the angiogenic protein analysis. The parameters, identical to those in Table 1, are compared in Supplementary Table S3. Here, the control group comprised three cases of GDM, while the SARS-CoV-2 infection group consisted of three cases of GDM and one case of pregnancy-induced hypertension.

## 2.2. Immunoblotting

The protein lysates and cell culture supernatant were prepared as described previously [35]. Protein samples of 20–25 µg were separated by protein gels (Bio-Rad, Hercules, CA, USA) and then transferred to nitrocellulose membranes (Bio-Rad, Hercules, CA, USA). The membranes were blocked in 5% nonfat milk at room temperature for 1 h, then incubated at 4 °C overnight, with primary and secondary antibodies listed in Supplementary Table S1. Blots were enhanced by Super Signal West Dura Extended Duration Substrate Kit (Thermo Fisher Scientific, Carlsbad, CA, USA). Images were captured using the ChemiDoc XRS+ system (Bio-Rad). Image J2 (Rawak Software Inc., Stuttgart, Germany) was employed for band intensity analysis. Protein expression levels were normalized to β-Actin expression. For inter-blot normalization, signal values were standardized to a consistent sample present on all blots.

## 2.3. Immunofluorescence and Immunohistochemical Examination

For immunofluorescence staining of 5 µm placenta sections, the detailed procedure is shown in Duan et al., 2021 [36]. The antibodies used are listed in Supplementary Table S2. The stained sections were visualized under a confocal fluorescence microscope (Leica SP5, Wetzlar, Germany) and photographed with LAS AF software version 4.0 (Leica, Wetzlar, Germany). A minimum of three placental tissues for each condition was investigated.

For immunohistochemical (IHC) staining, the tissue sections were subjected to routine dewaxing, antigen retrieval, and blocking. Subsequently, tissue sections were incubated in 3% $H_2O_2$ in methanol for 10 min to quench endogenous peroxidase activity. Primary antibodies were left to incubate overnight at 4 °C followed by a 1 h incubation with secondary antibodies at 37 °C. Antibodies are shown in Supplementary Table S2. Biotin and horseradish peroxidase-conjugated streptavidin were incubated to amplify the target antibody signal. Subsequently, the sections were counterstained with hematoxylin, dehydrated with gradient ethanol, purified with xylene, mounted, and observed under a ZEISS Axiophot microscope. The sections were finally photographed by a digital camera (DS-U1, Nikon, Tokyo, Japan) with NIS-Elements BR 3.0 software.

## 2.4. Statistical Analysis

In this study, the sample size was determined using G*Power 3.1 (Heinrich-Heine-University Düsseldorf, Düsseldorf, Germany). Statistical analysis utilized GraphPad Prism 8.0 (STATCON, Witzenhausen, Germany). The Shapiro–Wilk test assessed data normality. For normally distributed data, an independent *t*-test was used; non-normally distributed data underwent the Mann–Whitney test. Significance was set at $p \leq 0.05$.

## 3. Results

### 3.1. Characteristics of Pregnant Women and Neonatal Outcome with and without Maternal SARS-CoV-2 Infection in Term Pregnancies

Table 1 presents the clinical information of the pregnant women in the third trimester for the placenta analysis and the neonatal outcome with and without SARS-CoV-2 infection in term pregnancies. The patient cohort included 23 control and 24 SARS-CoV-2-positive pregnant women. No significant differences were observed in terms of maternal age, gestational age, cesarean section rate, neonatal birth weight, and Apgar score between the

two groups at delivery. Leukocyte levels showed a significant decrease, while C-reactive protein (CRP) was significantly increased in the SARS-CoV-2-positive group.

### 3.2. Serum sFlt-1, sFlt-1/PlGF, and sEng Were Significantly Elevated in Maternal Blood from Patients Infected with SARS-CoV-2

A total of 14 control and 14 SARS-CoV-2-infected blood samples of pregnant women were analyzed. Supplementary Table S3 shows the clinical characteristics of this patient cohort and the neonatal outcomes. At delivery, there were no notable distinctions in maternal age, gestational age, rate of cesarean section, neonatal birth weight, and Apgar score between the two groups. However, due to limitations in sample availability, the blood collection time for the control group was significantly earlier than that for the SARS-CoV-2 group. SARS-CoV-2-infected patients exhibited a significant decrease in leukocyte count compared to the control group, while CRP showed an increasing trend without statistical significance, which is somewhat consistent with the results in Table 1.

In maternal blood samples from SARS-CoV-2-positive patients, there was a significant elevation in the anti-angiogenic factors sFlt-1 ($p = 0.0005$) (Figure 1A) and sEng ($p = 0.0419$) (Figure 1D) along with a significant increase in the sFlt-1/PlGF ratio when compared to controls ($p < 0.0001$) (Figure 1C). PlGF, a member of the vascular endothelial growth factor family, displayed a significant decrease ($p = 0.0001$) (Figure 1B).

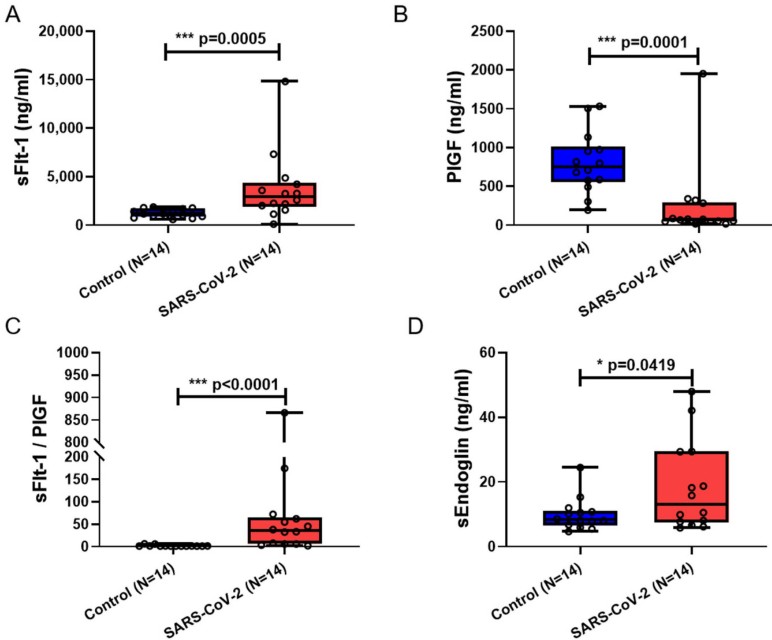

**Figure 1.** SARS-CoV-2 impact on maternal blood serum angiogenic biomarkers. The sFlt-1 (**A**), PlGF (**B**), sFlt-1/PlGF (**C**), and sEndoglin (**D**) levels in maternal blood sera of pregnant women infected with and without SARS-CoV-2. Data are represented as means ± SD; * $p < 0.1$, *** $p < 0.001$ significantly upregulated compared to control group.

### 3.3. Decreased Vascular Remodeling and Increased ROS Level in Placental Villi from Patients Infected with SARS-CoV-2

Figure 2 shows the representative histological findings of placental chorionic tissue obtained from pregnant women with or without SARS-CoV-2 infection, as determined by H&E staining. Figure 2A–D depicts placental villi obtained from two SARS-CoV-2-negative women, while Figure 2E–H represents villi from two SARS-CoV-2-positive patients. All placental tissues were from term pregnancies with comparable gestational ages. A comparative analysis of the images revealed that the placental stem villous arteries presented thickened vascular walls and narrowed lumens after SARS-CoV-2 infection, as already verified by Gychka et al. [28].

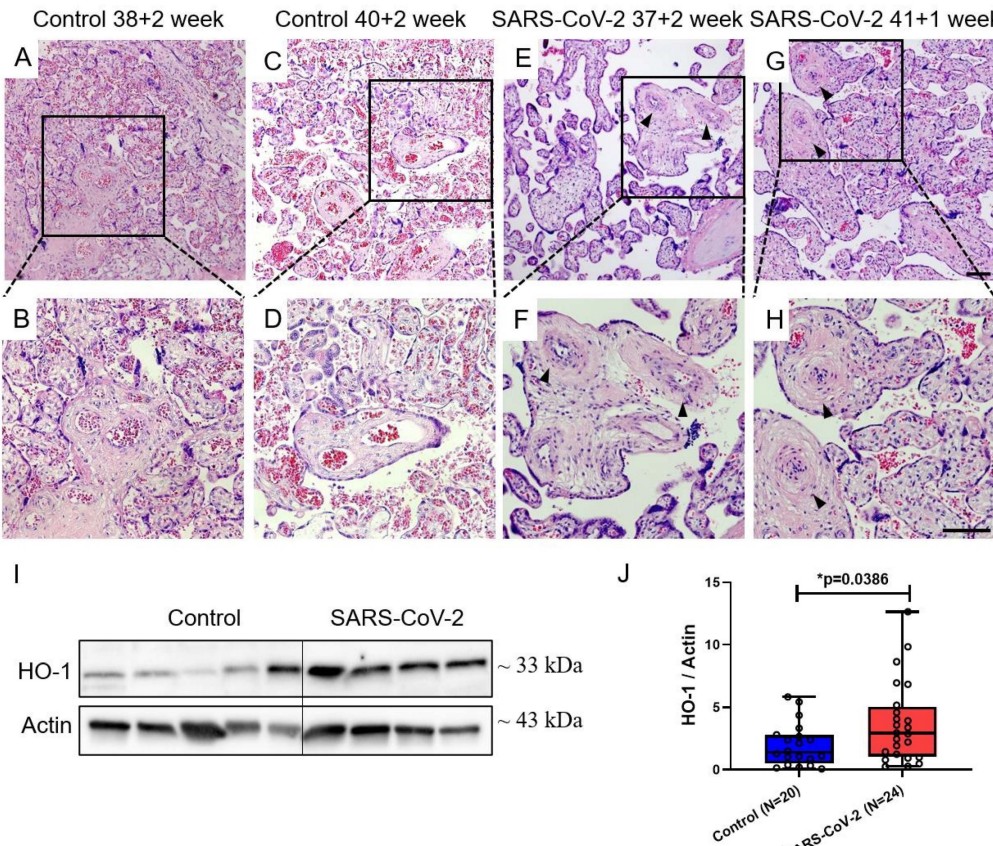

**Figure 2.** Vascular remodeling and HO-1 protein expression in placental villi of pregnant women infected with/without SARS-CoV-2. (**A–D**) H&E staining images of normal pregnant women's placental villi. (**E–H**) H&E staining images of SARS-CoV-2-positive patients' placental villi. Arrowheads represent remodeled vessels. (**I**) Representative immunoblot images of HO-1 protein expression in placental villi. (**J**) Statistical analysis of (**I**). (**B,D,F,H**) represent enlarged views of the areas shown by black boxes in images (**A,C,E,G**), respectively. Data are represented as means ± SD; * $p < 0.05$ significantly upregulated compared to control group. Scale bar: 100 μm.

As with other respiratory viral infections, SARS-CoV-2 can overall induce reactive oxygen species (ROS), as previously reported [37,38]. Heme oxygenase-1 (HO-1) is an enzyme that plays a critical role in heme metabolism and the cellular response to stress. It is encoded by the HMOX1 gene and is widely expressed in various tissues, including placenta [39]. The upregulation of HO-1 is associated with a wide range of physiological and pathological conditions, including inflammation and oxidative stress [40]. Furthermore, as previously shown by us, upon SARS-CoV-2 infection in pregnancy, the levels of cytokines IL-6, IL-18, IL-1β, and TNF-α within the serum of infected pregnant women were increased [41]. By measuring HO-1 protein levels in the placental tissue cohort, we verified significantly increased levels of HO-1 protein in placentas of SARS-CoV-2-positive patients compared to control ($p = 0.0386$) (Figure 2I,J), which implied that placentas in the SARS-CoV-2 group were exposed to higher levels of oxidative stress.

### 3.4. CCN1 and pNF-κB Expression Was Increased in Placental Villi from SARS-CoV-2-Infected Pregnancies

CCN1 is known for pro-inflammatory responses by recruiting immune cells [14,15]. Here, we found that placental CCN1 transcript and protein level were significantly elevated in patients infected with SARS-CoV-2 ($p = 0.0063$, $p = 0.0483$, respectively) (Figure 3A–C). Immunofluorescence staining showed that CCN1 was expressed in the cytoplasm and nucleus of the syncytiotrophoblast (STB), stained with cytokeratin 7 (CK7) as an STB marker, as well as in mesenchymal and endothelial cells in both patient groups. We observed a

stronger fluorescence intensity of CCN1 in placental villi of patients infected with SARS-CoV-2 compared to controls (Figure 3D–G). NF-κB is a downstream signaling factor of CCN1 promoting inflammatory responses [20,42]. Immunoblotting results revealed a significant elevation in the ratio of phospho-NF-κB to NF-κB in placentas from patients with SARS-CoV-2 infection ($p$ = 0.0145) (Figure 4A,B).

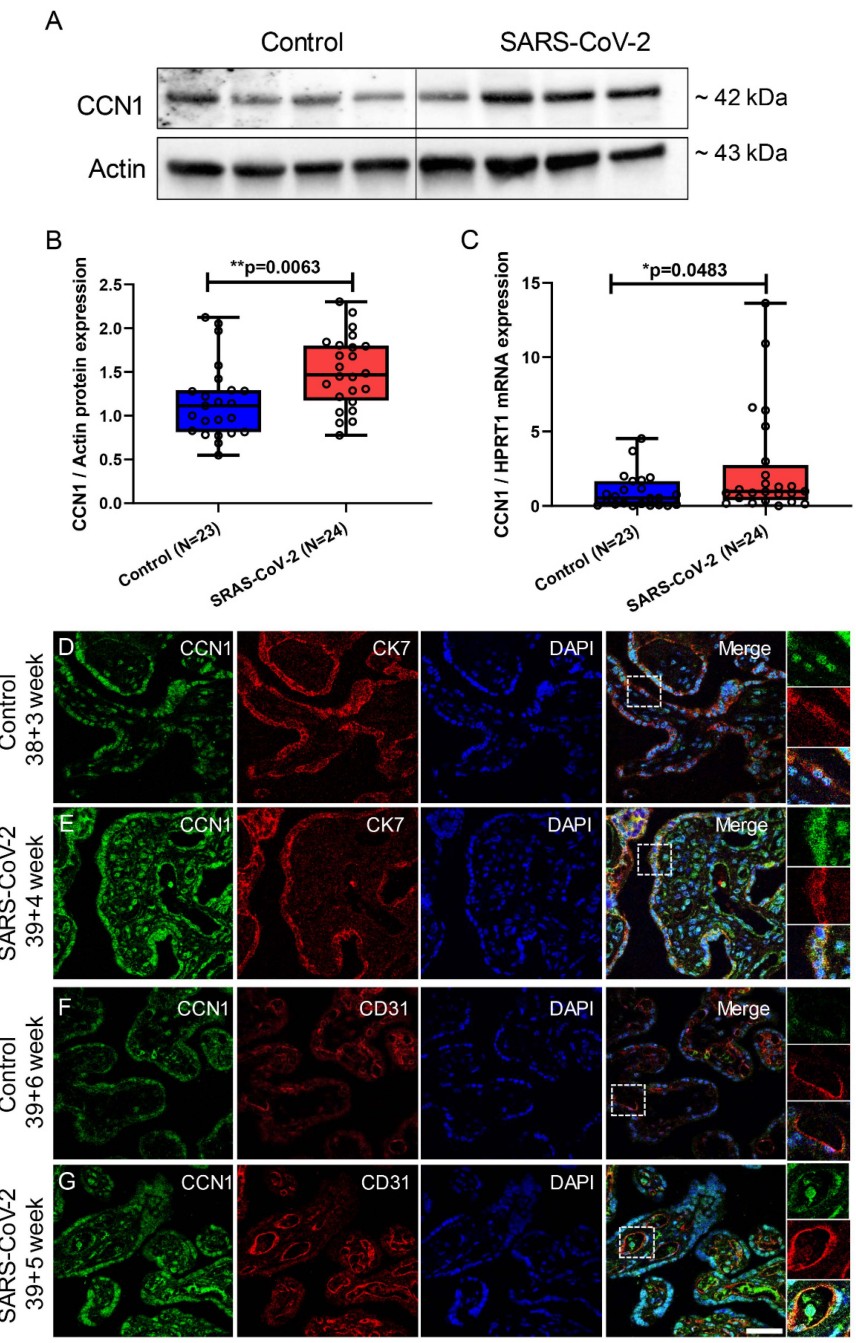

**Figure 3.** CCN1 expression in placental villi of pregnant women with (n = 24) and without (n = 23) SARS-CoV-2. (**A**) Representative immunoblot images of CCN1 protein expression in placental villi. (**B**) Statistical analysis of (**A**). (**C**) The relative CCN1 mRNA levels in placental villi. (**D,E**) Double immunofluorescence staining of CCN1 and the epithelial marker CK7 on placental villus tissue. (**F,G**) Double immunofluorescence staining of CCN1 and the endothelial cell marker CD31 on placental villus tissue. The white boxes in the panels of (**D–G**) on the right side are enlarged images of the white dotted boxes in the single fluorescent channel. Data are represented as means ± SD; * $p$ < 0.05, ** $p$ < 0.01 significantly upregulated compared to control group. Scale bar: 50 μm.

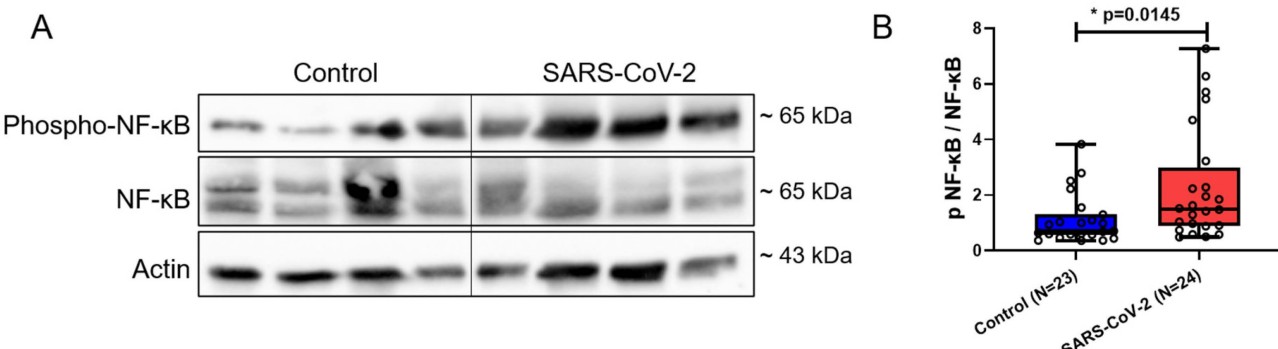

**Figure 4.** Phospho-NF-κB/NF-κB expression in placental villi of pregnant women with (n = 24) and without SARS-CoV-2 (n = 23). (**A**) Representative immunoblot images of phospho-NF-κB and NF-κB protein expression levels in placental villi. (**B**) Statistical analysis of (**A**) to represent the ratio of phospho-NF-κB to NF-κB. Data are represented as means ± SD; * $p < 0.05$ significantly upregulated compared to control group.

### 3.5. Flt-1 and Endoglin (Eng) Expression Was Increased in Placental Villi from SARS-CoV-2-Infected Pregnancies

As shown above, elevated levels of serum sFlt-1 and sEng were observed in pregnant women infected with SARS-CoV-2 (Figure 1). During pregnancy, sFlt-1 and sEng primarily originate from the synthesis of membrane-bound forms of Flt-1 (VEGF receptor-1) and Eng by placental trophoblasts, which are subsequently cleaved at the membrane by proteolytic processes and modified into secretory proteins, which are released into the bloodstream [23,43]. Therefore, we assessed the protein levels of Flt-1 and Eng in placental villi obtained from SARS-CoV-2-positive and -negative women, as determined by immunoblotting (Figure 5A). A significant increase in Flt-1 protein level was observed in placentas from SARS-CoV-2-positive patients ($p = 0.0180$) (Figure 5B). Additionally, elevated levels of Eng protein in placental villi from SARS-CoV-2-positive women were detected ($p = 0.0044$) (Figure 5C). Figure 5D,E shows the localization of Flt-1 in the placenta by immunofluorescence staining. The expression of Flt-1 was detected in both STB and stromal cells in addition to endothelial cells as confirmed by its co-localization with CD31, a marker for endothelial cells. Notably, the SARS-CoV-2 group exhibited enhanced Flt-1 staining primarily in STBs and stromal cells, with strong fluorescence intensity observed, while Flt-1 in endothelial cells showed an intensive fluorescence intensity in both groups (Figure 5D,E). In contrast, Eng protein expression was detected exclusively in STB, as confirmed by its co-localization with the STB marker CK7, and, here, located in the apical membrane (Figure 5F,G). The immunofluorescent staining for Flt-1/CK7 and Eng/CD31 is illustrated in Supplementary Figure S1. In conclusion, the placental villi obtained from SARS-CoV-2-positive patients exhibited higher levels of Eng and Flt-1 protein as compared to the control group.

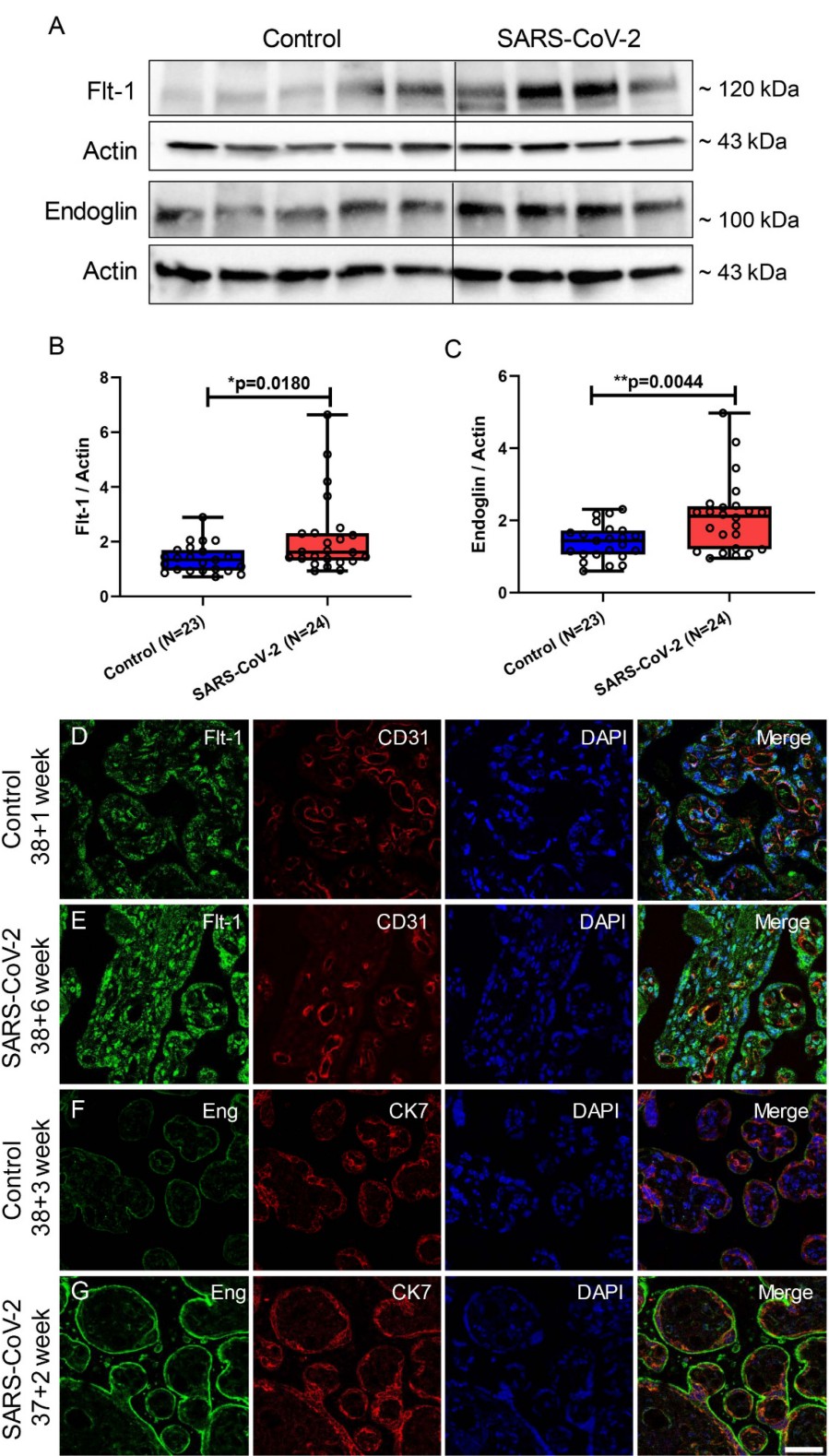

**Figure 5.** Flt-1/endoglin expression in placental villi of pregnant women with (n = 24) and without (n = 23) SARS-CoV-2. (**A**) Representative immunoblot images of Flt-1 and Eng protein expression in placental villi. (**B**,**C**) Statistical analysis of (**A**). (**D**,**E**) Double immunofluorescence staining of Flt-1 and CD31 on placental villus tissue. (**F**,**G**) Double immunofluorescence staining of Eng and CK7 on placental villus tissue. Data are represented as means ± SD; * $p < 0.05$, ** $p < 0.01$ significantly upregulated compared to control group. Scale bar: 50 μm.

### 3.6. TGFβR1 and eNOS Expression Are Counterregulated in Placental Villi from SARS-CoV-2-Infected Pregnancies

TGFβR1 is a specific receptor of TGFβ and can regulate nitric oxide (NO) release by activating eNOS signaling after binding, thereby participating in spiral artery remodeling during pregnancy [44,45]. However, sEng can competitively bind to the TGFβR1, leading to placental vascular dysfunction [44]. In our study, we found elevated serum levels of sEng in SARS-CoV-2-positive patients (Figure 1C), which may result in enhanced sEng and TGFβR1 binding and reduced Eng and TGFβ binding. In contrast, TGFβR1 expression was found to be significantly increased in placentas from SARS-CoV-2-positive patients ($p = 0.0339$) (Figure 6A–C), whereas eNOS showed a significant reduction in the SARS-CoV-2-positive group ($p = 0.0147$) (Figure 6A,B,D).

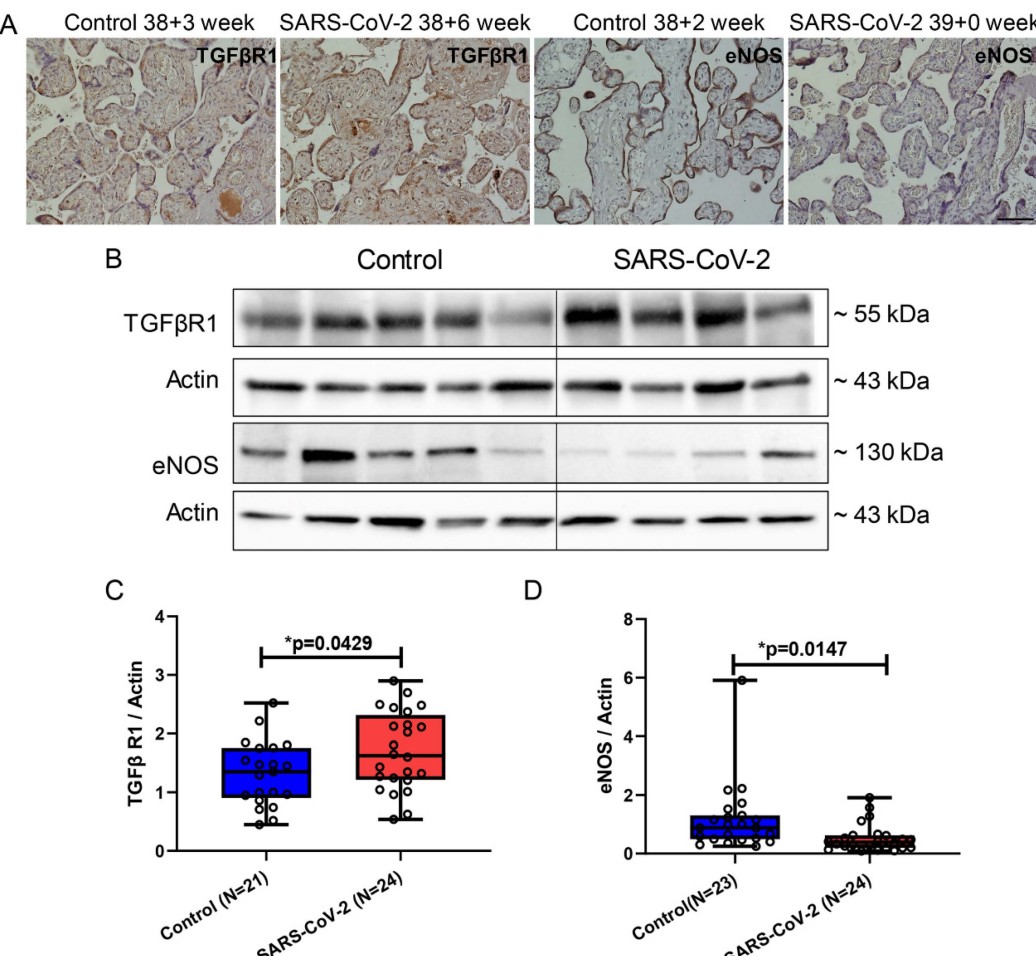

**Figure 6.** TGFβR1 and eNOS protein localization in placental villi of pregnant women with and without SARS-CoV-2. (**A**) Immunohistochemical staining images of TGFβR1 and eNOS localization in placental villi. (**B**) Representative immunoblot images of TGFβR1 and eNOS protein expression in placental villi. (**C,D**) Statistical analysis of (**B**). Data are represented as means ± SD; * $p < 0.05$ significantly upregulated or downregulated compared to control group. Scale bar: 100 μm.

### 3.7. Recombinant CCN1 Induced Flt-1 and Endoglin Protein Expression in SGHPL-5 Trophoblast Cells, along with the Release of sFlt-1 and sEng

The study on placental tissues provided valuable insights into the potential mechanisms underlying placental changes induced by SARS-CoV-2 infection, including the involvement of CCN1-mediated inflammatory responses. However, further mechanistic studies, such as in vitro experiments using placental cell lines, can elucidate the precise molecular pathways involved and help validate the observed associations.

A previous study by us showed that CCN1 plays a key role in regulating trophoblast differentiation [46]. We revealed that treating SGHPL-5 trophoblast cells with CCN1 resulted in a reduced cell proliferation by inducing cell cycle arrest and elevating p21 expression as well as an increase in senescence and migration properties by the upregulation of the pAkt and pFAK kinases. In order to investigate the effects of CCN1 on the level of angiogenic proteins in trophoblast cells in vitro, SGHPL-5 trophoblast cells were incubated with CCN1 protein. We observed that CCN1 was able to induce the expression of membranous Flt-1 and Eng proteins in SGHPL-5 cells (Figure 7A–C). Furthermore, the levels of sFlt-1 and sEng in the cell culture supernatant were also elevated, showing that CCN1 could also increase the secretion of sFlt-1 and sEng (Figure 7D–F).

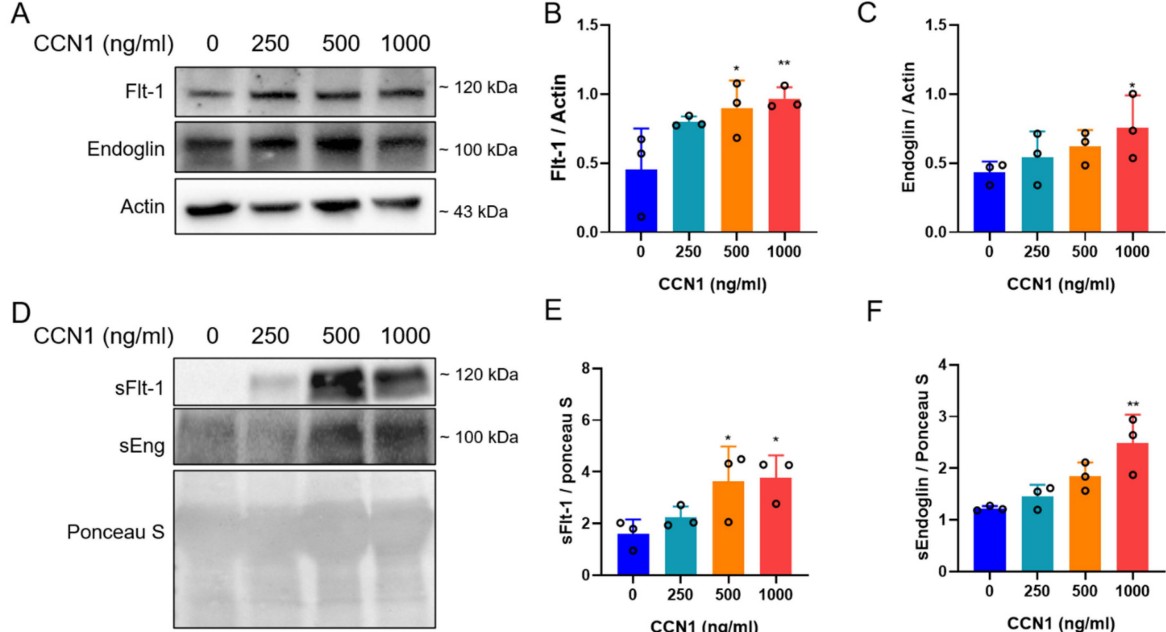

**Figure 7.** Recombinant CCN1 induced Flt-1 and endoglin expression and sFlt-1 and sEng secretion in SGHPL-5 cells. (**A**) Effects of different concentrations of CCN1 on the expression level of Flt-1 and Eng in SGHPL-5 cells. (**B,C**) Statistical analysis of (**A**) to represent Flt-1 and Eng protein levels. (**D**) Effects of different concentrations of CCN1 on the secretion level of sFlt-1 and sEng in SGHPL-5 supernatant. (**E,F**) Statistical analysis of (**D**) to represent Flt-1 and Eng protein levels. Each experiment was repeated three times. Data are represented as means $\pm$ SD; * $p < 0.05$, ** $p < 0.01$ significantly upregulated compared to control group.

## 4. Discussion

This study verified previous findings that term pregnant women infected with SARS-CoV-2 exhibited histopathological changes in placental vascular remodeling. The new finding here is that CCN1 may induce this elevation of anti-angiogenic factors such as sFlt-1 and sEng in the serum of SARS-CoV-2-positive patients. This is strengthened by the observation of a significant upregulation of the vascular inflammation-promoting protein CCN1 and its downstream inflammatory factor phospho-NF-κB/NF-κB in the placenta villous tissues of pregnant women infected with SARS-CoV-2 at term. These molecular changes may potentially serve as an underlying cause of the observed histopathological alterations in placental vascularization.

Our study cohort consisted of 23 placentas from normal pregnant women and 24 placentas from patients with SARS-CoV-2. Although, as a limiting factor, there were six cases in the control group and seven cases in the SARS-CoV-2 group with coexisting comorbidities, it is noteworthy that more than one-third of the placental samples were obtained from pregnancies without any other gestational syndrome. All placental samples used in this study were collected from term pregnancies, and the patients infected with SARS-

CoV-2 delivered within 1–14 days after infection. We therefore focused, in this study, on the short-term effects of SARS-CoV-2 infection on full-term maternal and fetal outcomes. Pathological examination of the placental tissues from term pregnant women infected with SARS-CoV-2 revealed significant vascular alterations, characterized by thickened vessel walls and narrowed lumens. These findings are consistent with the results reported by Gychka et al., who also identified substantial vascular wall thickening and lumen narrowing in the placentas of SARS-CoV-2-positive patients in the second as well as third trimester [28]. Hence, the key question arises: what is the direct causative factor underlying these modifications? Giardini et al. suggested that SARS-CoV-2 infection shares similar pathological alterations with preeclampsia, indicating varying degrees of endothelial dysfunction under both conditions [47]. As previously stated, sFlt-1 and sEng serve as anti-angiogenic markers for this disease, widely used in clinical assessment. Moreover, Negro et al. already demonstrated significantly elevated serum levels of sFlt-1 in deceased, non-pregnant patients with severe COVID-19 compared to survivors [48], which could be confirmed by our study for pregnant patients infected with SARS-CoV-2 with elevated serum levels of anti-angiogenic molecules sFlt-1 and sEng already seen after asymptomatic or mild SARS-CoV-2 infection compared with normal pregnant women. Unfortunately, due to an insufficient number of cases, we could not discriminate into categories of mild and asymptomatic pathology in our study.

Flt-1 is widely expressed in various *human* tissues, including liver, prostate, and placenta [49], while Eng is expressed in tissues such as heart, kidney, lung, and placenta [50]. In preeclampsia patients, circulating sFlt-1 and sEng are primarily synthesized and secreted by placental villi [23,24]. Considering our study population consisting of pregnant women, we proceeded to examine the expression of membranous Flt-1 and Eng receptors in the placental villi. We observed a significant upregulation of membrane Flt-1 and Eng expression in the placentas of SARS-CoV-2-positive patients, providing a possible explanation for the elevated serum levels of sFlt-1 and sEng in SARS-CoV-2-positive patients. Excess sFlt-1 blocks the binding of PlGF and VEGF to their receptors, leading to endothelial dysfunction and persistent vasoconstriction in the patients. Under normal circumstances, the binding of TGFβ and TGFβR1 facilitates the release of eNOS, which plays a crucial role in the production of NO for maintaining vascular dilation and homeostasis [51]. We speculate that, in the context of SARS-CoV-2 infection, an excessive amount of sEng binds to TGFβ, thereby obstructing the TGFβ signaling pathway. Consequently, compensatory upregulation of TGFβR1 occurs in the placenta, as found in this study. Nevertheless, this upregulation is insufficient to sustain the normal flow of the TGFβ pathway, ultimately leading to reduced eNOS release. Consequently, there is a diminished generation of nitric oxide and impaired vascular dilation. In summary, we revealed a possible link that SARS-CoV-2 infection increases the risk of sustained vasoconstriction in the placental vasculature.

The sFlt-1/PlGF ratio serves as a biomarker for oxidative stress in trophoblast cells and endothelial cells. A meta-analysis by Kosinska-Kaczynska et al. showed that the sFlt-1/PlGF ratio significantly increased after SARS-CoV-2 infection and there was no significant difference between symptomatic and asymptomatic pregnant women with SARS-CoV-2 infection [52]. Despite 62.5% of SARS-CoV-2-positive patients being asymptomatic and 37.5% exhibiting mild symptoms in our study population, we also observed a higher sFlt-1/PlGF ratio in the SARS-CoV-2 group compared to the control group, consistent with the conclusion of Kosinska-Kaczynska et al. In addition, they proposed that patients with severe COVID-19 had a higher sFlt-1/PlGF ratio than those with non-severe COVID-19.

Additionally, following infection with SARS-CoV-2, the onset of a cytokine storm occurs in pregnant women as well as in placentas, as previously mentioned by us [41]. The accumulation of neutrophils and macrophages in the placental stroma of SARS-CoV-2-infected pregnant women was reported by multiple studies, indicating the induction of a systemic inflammatory response by SARS-CoV-2 [30,53]. Consequently, we aimed to investigate whether SARS-CoV-2 infection also triggers an inflammatory response in the placenta.

CCN1, recognized as a pro-inflammatory cytokine involved in regulating inflammatory molecules, is known to exhibit upregulated expression in response to viral or bacterial infections. In the context of normal pregnancy, CCN1 has been observed by us and others to be expressed in extravillous trophoblast (EVT) cells, STB, endothelial cells, and stromal cells within the placenta of normal pregnancies [54,55]. Consistent with findings from other researchers, our study revealed an elevation in CCN1 expression in placental villous tissue after the infection of SARS-CoV-2. On the one hand, CCN1 can activate the NF-κB signaling pathway within trophoblast SGHPL-5 cells by binding to unknown cell surface receptors, which may lead to increased transcription of inflammation-related genes such as cytokines IL-1β, IL-6, TNF-α, etc. On the other hand, CCN1 can facilitate the infiltration and activation of inflammatory cells. CCN1 promotes the migration and activation of neutrophils, monocytes, and macrophages through integrin CD11b and cell surface heparan sulfate proteoglycan syndecan-4 [15,21]. Previous reports highlighted the upregulation of CCN1 in the small intestine in *humans* following SARS-CoV-2 infection [18]. Similarly, our study found that SARS-CoV-2 may increase the expression of CCN1 in the placenta. It was demonstrated that SARS-CoV-2 triggers a systemic inflammatory response, including inflammation in the placenta [56]. CCN1, in turn, can induce and activate various immune cells, leading to the generation and secretion of cytokines [57]. Hence, we hypothesize that the overexpression of CCN1 in the placenta is a crucial factor in initiating placental inflammation. The upregulation of inflammatory cytokines, such as TNF-α and IL-6, causes endothelial dysfunction, manifested by the increased production of adhesion molecules, enhanced vascular permeability, and promotion of endothelial cell proliferation and migration of vascular smooth muscle cells [58]. Our investigation indicated an elevation in placental HO-1 protein, a marker associated with antioxidant response, among individuals afflicted with SARS-CoV-2 infection, suggesting a potential increase in oxidative stress. We assume that SARS-CoV-2 infection in term pregnancy induces oxidative stress and inflammation upon CCN1 and other cytokines and further enhances the generation and secretion of the anti-angiogenic factors sFlt-1 and sEng, ultimately leading to placental vascular dysfunction and remodeling. Although the mechanism remains elusive, our in vitro study substantiated that CCN1 can induce trophoblast cells to express and secrete sFlt-1 and sEng, thereby strengthening this hypothesis. In former studies, we already showed that CCN1 regulated trophoblast differentiation in SGHPL-5 cells by inducing cell cycle arrest, activating Akt and FAK signaling, and inducing senescence, which may be associated to the increase in anti-angiogenic molecules. Interestingly, there are similarities between SARS-CoV-2 infection and preeclampsia in terms of dysregulated vascular growth factors and vascular dysfunction. However, it is important to note that SARS-CoV-2 infection is characterized by acute inflammation, while the pathogenesis of preeclampsia involves chronic immune activation resulting in dysregulation of both regulatory and inflammatory cytokines. The specific mechanisms underlying these two different pathological conditions require further research and elucidation.

The schematic overview (Figure 8) summarizes the identified CCN1-mediated signaling pathways in term placentas of SARS-CoV-2-infected women in the third trimester. After SARS-CoV-2 infection in the third trimester of pregnancy, ROS increased and HO-1 expression was elevated, which activated immune cells and triggered a systemic cytokine storm. These cytokines entered the placental tissue through the blood circulation. In placental tissues, SARS-CoV-2 infection upregulated the inflammatory factor CCN1 and activated its downstream signaling factor NF-κB. Activation of this signaling pathway might further recruit monocytes, macrophages, and neutrophils to the placental tissue, leading to the secretion of cytokines. As our previous research found, the serum levels of cytokines IL-6, IL-10, IL-1β, and TNF-α in pregnant women infected with SARS-CoV-2 were significantly increased [41]. Also, Shook et al. found a variety of cytokines such as IL-1, IL-8, TNFα, and TGFβ also showed an elevated trend after SARS-CoV-2 infection in pregnant women [56]. Excessive cytokines may induce the synthesis and secretion of

anti-angiogenic factors such as sFlt1-1 and sEng in the placenta, ultimately resulting in placental endothelial dysfunction and vascular fibrosis.

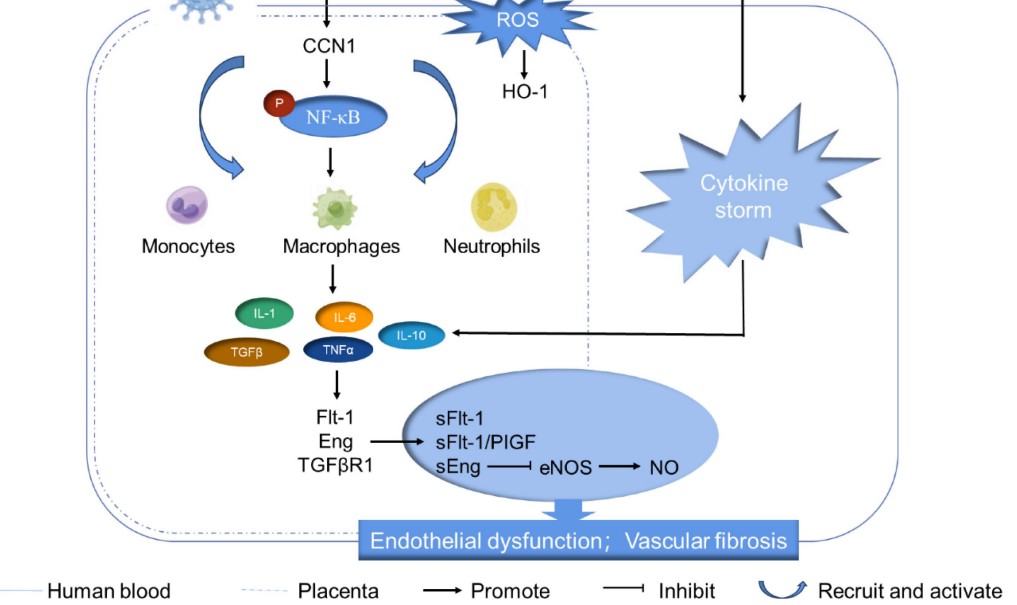

**Figure 8.** Schematic diagram of CCN1-mediated signaling pathway in pregnant women with SARS-CoV-2-infection. Upon SARS-CoV-2 infection, ROS increase rapidly, triggering immune cell activation and instigating a systemic cytokine storm. These cytokines enter the placental tissue through the blood circulation. In the placenta, SARS-CoV-2 upregulates the inflammatory factor CCN1 and activates its downstream effector NF-κB. This signaling cascade may subsequently recruit monocytes, macrophages, and lymphocytes to the placental tissue, leading to additional cytokine secretion. The excessive cytokines may induce the synthesis and release of abundant anti-angiogenic factors within the placenta, ultimately causing disruption of endothelial function and vascular fibrosis.

Our study revealed the potential of the CCN1-mediated signaling pathway as a therapeutic target for improving placental function after SARS-CoV-2 infection in term pregnancy. Furthermore, integrating these biomarkers such as sFlt-1, PlGF, and sEng into clinical practice can be helpful in assessing and monitoring the risk in pregnant women infected with SARS-CoV-2.

Study limitations: The SARS-CoV-2 cohort lacked severe cases due to limited samples; asymptomatic and mild cases are more common clinically. Furthermore, longitudinal assessments including serial measurements throughout pregnancy and follow-up analyses postpartum were not conducted, which could provide valuable insights into the temporal dynamics of placental changes and their implications for maternal and neonatal outcomes. It is likely that these changes may be more dangerous in the second and early third trimesters and may explain the observed increased risk of preterm birth and intrauterine fetal death, as demonstrated in the prospective german register study COVID-19 Related Obstetrics and Neonatal Outcome Study including 8032 pregnant women [59]. Moreover, while the role of CCN1 and its downstream pathway was emphasized, further exploration using placental cell lines is needed for the intricate interaction of inflammatory cells and cytokines within placental tissues to unravel the whole signaling mechanism. Future prospective trials and mechanistic studies that address the above limitations are necessary.

## 5. Conclusions

Our study revealed that elevated levels of CCN1 in the placentas of pregnant women infected with SARS-CoV-2 played a significant role in activating inflammatory responses.

This activation triggered the production of anti-angiogenic factors such as sFlt-1 and sEng, potentially leading to abnormal placental vascular architecture and exacerbating placental dysfunction associated with SARS-CoV-2 infection.

**Supplementary Materials:** The following supporting information can be downloaded at: https://www.mdpi.com/article/10.3390/cimb46040221/s1. Figure S1: Flt-1 and Endoglin were increased in placental villous tissue of SARS-CoV-2 infected women; Table S1: Antibodies used for immunoblotting; Table S2: Antibodies used for immunohistochemical/immune-fluorescence staining; Table S3: SARS-CoV-2 infected vs. non-infected pregnancies: patient cohort for angiogenic protein analysis. Refs. [36,60–62] are cited in the Supplementary Materials.

**Author Contributions:** Conception/design: Y.M., L.D., A.G. and A.I. Resources: R.K. Investigation/data analysis: Y.M., L.D., B.R., A.G. and A.I. Manuscript drafting: Y.M., L.D., B.R., A.G. and A.I. All authors have read and agreed to the published version of the manuscript.

**Funding:** We thank the Chinese Scholarship Council (CSC) (202108130056) for giving funding to support the MD study of Yuyang Ma and acknowledge support by the Open Access Publication Fund of the University of Duisburg-Essen, Essen, Germany.

**Institutional Review Board Statement:** Authors confirm adherence to the journal's ethical policies, obtaining necessary committee approval and complying with the local ethics committee of the University of Duisburg-Essen, Essen, Germany (12-5212-BO, 21-10462-BO).

**Informed Consent Statement:** Informed consent was obtained from all subjects involved in the study.

**Data Availability Statement:** Upon reasonable request, the corresponding author can provide access to the datasets utilized and analyzed in the current study.

**Acknowledgments:** We extend appreciation to lab technicians Gabriele Sehn, Elisa Marie Elfroth, Jens Rasch, Sieglinde Arndt, and Ute Kirsch, also to nurses who collected the patient samples at University Hospital Essen. Additionally, we are indebted to all participants.

**Conflicts of Interest:** The authors declare no conflicts of interest.

## Abbreviations

BMI, body mass index; CK7, cytokeratin 7; COVID-19, coronavirus disease 2019; CRP, C-reactive protein; CYR61/CCN1, cysteine-rich protein 61; eNOS, endothelial nitric oxide synthase; EVT, extravillous trophoblast; FGR, fetal growth restriction; GDM, gestational diabetes mellitus; HO-1, heme oxygenase-1; IHC, immunohistochemical; MVM, maternal vascular malperfusion; NO, nitric oxide; PlGF, placental growth factor; ROS, reactive oxygen species; SARS-CoV-2, severe acute respiratory syndrome coronavirus 2; sEng, soluble endoglin; sFlt-1, soluble fms-like tyrosine kinase 1; STB, syncytiotrophoblast; TGFβ, transforming growth factor beta; TGFβR, transforming growth factor beta receptor; VEGF, vascular endothelial growth factor; VEGFR-1/Flt-1, vascular endothelial growth factor receptor-1; VEGFR-2, vascular endothelial growth factor receptor-2.

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
