# Peer review of "CCN1-Mediated Signaling in Placental Villous Tissues after SARS-CoV-2 Infection in Term Pregnant Women: Implications for Dysregulated Angiogenesis"

_cimb, doi:10.3390/cimb46040221_

Round 1

Reviewer 1 Report

Comments and Suggestions for Authors

I consider this article wonderful. I would just like to highlight one factor that I consider important. In fact, although I believe that the entire metabolic cascade of cytokine activation is well documented, I have some reservations about the positivity of newborns for SARS-Cov 2. Delivery time < 14 days from maternal infection may not be sufficient to demonstrate the negativity of the newborn, because a longer viral replication time may be required to reach a PCR detection cut-off. Furthermore, in my opinion, the throat swab for newborns may not be the best place for testing, knowing that the fetus does not breathe room air. There is data on detecting the virus from cord blood and this could be the main gateway to active infection, so looking for the virus in meconium and neonatal urine could be a significant step. I suggest reading this article to complete the references "McMahon CL, Castro J, Silvas J, Muniz Perez A, Estrada M, Carrion R Jr, Hsieh J. Fetal brain vulnerability to SARS-CoV-2 infection. Brain Behav Immun. 2023 Aug;112:188-205. doi: 10.1016/j.bbi.2023.06.015. Epub 2023 Jun 16. PMID: 37329995; PMCID: PMC10270733."

Author Response

Comment: I consider this article wonderful. I would just like to highlight one factor that I consider important. In fact, although I believe that the entire metabolic cascade of cytokine activation is well documented, I have some reservations about the positivity of newborns for SARS-Cov 2. Delivery time < 14 days from maternal infection may not be sufficient to demonstrate the negativity of the newborn, because a longer viral replication time may be required to reach a PCR detection cut-off. Furthermore, in my opinion, the throat swab for newborns may not be the best place for testing, knowing that the fetus does not breathe room air. There is data on detecting the virus from cord blood and this could be the main gateway to active infection, so looking for the virus in meconium and neonatal urine could be a significant step. I suggest reading this article to complete the references "McMahon CL, Castro J, Silvas J, Muniz Perez A, Estrada M, Carrion R Jr, Hsieh J. Fetal brain vulnerability to SARS-CoV-2 infection. Brain Behav Immun. 2023 Aug;112:188-205. doi: 10.1016/j.bbi.2023.06.015. Epub 2023 Jun 16. PMID: 37329995; PMCID: PMC10270733."

Reply: Thank you for your insightful and positive feedback on our manuscript.

Indeed, the timeframe for testing newborns born < 14 days from maternal infection may not be sufficient to reliably demonstrate their negativity due to potential ongoing viral replication. Furthermore, your point regarding the choice of sampling site for testing, especially considering the unique respiratory physiology of fetuses, is noteworthy. However, the testing was routinely done by the Institute of Virology at our University Hospital according a standard procedure using a nasopharyngeal swab for all patients/newborns. Exploring alternative sampling methods such as meconium and neonatal urine could indeed may provide valuable insights into viral transmission and infection dynamics in newborns. We appreciate your suggestion to refer to the article " Fetal brain vulnerability to SARS-CoV-2 infection " by McMahon et al. and cite this article and its conclusions in the introduction part about the impact of SARS-CoV-2 on the fetus (Page 1, line 42).

Reviewer 2 Report

Comments and Suggestions for Authors

This study examines how placental pathology and angiogenic factor levels are affected by SARS-CoV-2 infection in term pregnant women, with a particular emphasis on the function of the inflammatory protein CYR61/CCN1. According to the research, SARS-CoV-2 infection causes significant vascular changes in the placental villi, which may be a factor in unfavorable pregnancy outcomes such preterm birth and eclampsia. Placental abnormalities are linked to increased CCN1 and pNF-κB, whereas elevated levels of anti-angiogenic factors sFlt-1 and sEng are identified as potential mediators of these pathological changes. Nonetheless, a number of the study's findings demand more attention and research.

The study mentions a retrospective single-center design, but it lacks details on the sample size and selection criteria. How were the infected and uninfected pregnant women identified and recruited into the study? Was there a matching process to ensure comparability between the groups?

Introducing color to the figures presented in the study could enhance the clarity and interpretation of the results, particularly for readers who may benefit from visual differentiation of key elements.

The study provides insights into the potential mechanisms underlying placental alterations induced by SARS-CoV-2 infection, including the involvement of CCN1-mediated inflammatory responses. However, further mechanistic studies, such as in vitro experiments using placental cell lines or animal models, could elucidate the precise molecular pathways involved and help validate the observed associations. This should be clarified in the manuscript.

While the study highlights the potential role of CCN1 and anti-angiogenic factors in SARS-CoV-2-associated placental pathology, the clinical implications of these findings remain to be fully elucidated. How might these molecular alterations inform clinical management and treatment strategies for pregnant women with SARS-CoV-2 infection?

The study primarily focuses on placental pathology and angiogenic factor levels at a single time point during term pregnancy. Longitudinal assessments, including serial measurements throughout pregnancy and follow-up postpartum, could provide valuable insights into the temporal dynamics of placental changes and their implications for maternal and neonatal outcomes. This should be clarified in the manuscript.

Finally, the work poses significant queries about the possible effects of SARS-CoV-2 infection on the health of expectant mothers and newborns. In what ways might the results of this study be applied to clinical practice to enhance the monitoring, risk assessment, and treatment of SARS-CoV-2-infected pregnant women?

Overall, even though the study clarifies the intricate relationship between placental pathology and SARS-CoV-2 infection, more investigation is required to support and build on these findings. To improve our knowledge of the effects of SARS-CoV-2 infection on mothers and fetuses and to guide therapeutic practice, it will be essential to address the aforementioned limitations by carefully planned prospective trials and mechanistic investigations. Information should be introduced in the text

Comments on the Quality of English Language

English is poor, there are quite a few syntax and grammar errors in the text

Author Response

This study examines how placental pathology and angiogenic factor levels are affected by SARS-CoV-2 infection in term pregnant women, with a particular emphasis on the function of the inflammatory protein CYR61/CCN1. According to the research, SARS-CoV-2 infection causes significant vascular changes in the placental villi, which may be a factor in unfavorable pregnancy outcomes such preterm birth and eclampsia. Placental abnormalities are linked to increased CCN1 and pNF-κB, whereas elevated levels of anti-angiogenic factors sFlt-1 and sEng are identified as potential mediators of these pathological changes. Nonetheless, a number of the study's findings demand more attention and research.

We thank the reviewer for the valuable comments to improve our manuscript.

The study mentions a retrospective single-center design, but it lacks details on the sample size and selection criteria. How were the infected and uninfected pregnant women identified and recruited into the study? Was there a matching process to ensure comparability between the groups?

We thank the reviewer for this critical question. The sample size was determined using G Power software in advance to ensure adequate statistical power to address the research question. The effect size (dz) was estimated to be 0.8 (according to previous studies such as PMID: 36121927, PMID: 35475405, PMID: 34974633), with a significance level (α) of 0.05 and a desired power (1-β) of 0.95. Based on these parameters, the minimum required total sample size was calculated to be n=23. Subsequently, 23 control pregnant women and 24 SARS-CoV-2 positive patients were included in the study. This information was added in materials and methods – chapter statistical analysis (Page 4, line 184).

The SARS-CoV-2 positive patients were diagnosed using nasopharyngeal RT-PCR at the Institute of Virology in our hospital. Pregnant patients were categorized according to the severity of their COVID-19 symptoms, following the WHO classification (PMID: 35917393), into asymptomatic/mildly or severely symptomatic groups. Control patients were selected to match the gestational age of the SARS-CoV-2 positive patients to ensure comparability between the groups. This has been stated in materials and methods – chapter study population (Page 3, line 138-146).

Introducing color to the figures presented in the study could enhance the clarity and interpretation of the results, particularly for readers who may benefit from visual differentiation of key elements.

Thank you for the valuable advice. Based on your suggestion, we changed all the figures according to use colored columns to improve the clarity and interpretability of the results.

The study provides insights into the potential mechanisms underlying placental alterations induced by SARS-CoV-2 infection, including the involvement of CCN1-mediated inflammatory responses. However, further mechanistic studies, such as in vitro experiments using placental cell lines or animal models, could elucidate the precise molecular pathways involved and help validate the observed associations. This should be clarified in the manuscript.

Thanks for the helpful suggestion. We are fully aware that our study results are just one piece of puzzle in the signaling cascade mediated by CCN1 and further detailed mechanistic studies using cell lines are needed to fully unravel the whole mechanism. We clarified this in the Result part (Page 12, line 345-349) as well as the discussion part, in limitations of the study (Page 16, line 522-525).

While the study highlights the potential role of CCN1 and anti-angiogenic factors in SARS-CoV-2-associated placental pathology, the clinical implications of these findings remain to be fully elucidated. How might these molecular alterations inform clinical management and treatment strategies for pregnant women with SARS-CoV-2 infection?

We thank the reviewer for this interesting question. We have addressed this concern by incorporating the potential clinical implications of our findings in the discussion section (Page 15, line 506-510). Specifically, our study revealed the potential of the CCN1-mediated signaling pathway as a possible therapeutic target for alleviating placental dysfunction associated with SARS-CoV-2 infection. Furthermore, integrating these biomarkers (sFlt-1, PlGF, sFlt-1/PlGF and sEng) into clinical practice can be used in assessing and monitoring the risk in pregnant women infected with SARS-CoV-2.

The study primarily focuses on placental pathology and angiogenic factor levels at a single time point during term pregnancy. Longitudinal assessments, including serial measurements throughout pregnancy and follow-up postpartum, could provide valuable insights into the temporal dynamics of placental changes and their implications for maternal and neonatal outcomes. This should be clarified in the manuscript.

We completely agree with the reviewers comment. However, our main focus was primarily on analyzing placental pathology and angiogenic factor levels at a single time point in term pregnancy. However, your suggestion regarding longitudinal assessments sounds promising. Conducting serial measurements and follow-up postpartum throughout the entire pregnancy would provide valuable insights into the temporal dynamics of placental changes and their implications for maternal and neonatal outcomes. We clarified this point in the discussion part-Study limitations (Page 15, line 512-522) and consider your suggestion for future research to explore these aspects more comprehensively.

Finally, the work poses significant queries about the possible effects of SARS-CoV-2 infection on the health of expectant mothers and newborns. In what ways might the results of this study be applied to clinical practice to enhance the monitoring, risk assessment, and treatment of SARS-CoV-2-infected pregnant women?

The potential clinical applications of our findings have been expanded in the discussion section (Page 15, line 506-510). In detail, our study revealed the potential of the CCN1-mediated signaling pathway as a therapeutic target for improving placental function after SARS-CoV-2 infection in term pregnancy. Furthermore, integrating these biomarkers such as sFlt-1, PlGF, and sEng into clinical practice can be helpful in assessing and monitoring the risk in pregnant women infected with SARS-CoV-2.

Overall, even though the study clarifies the intricate relationship between placental pathology and SARS-CoV-2 infection, more investigation is required to support and build on these findings. To improve our knowledge of the effects of SARS-CoV-2 infection on mothers and fetuses and to guide therapeutic practice, it will be essential to address the aforementioned limitations by carefully planned prospective trials and mechanistic investigations. Information should be introduced in the text

Yes we agree. We clarified this information in the discussion part-study limitations (Page 15, line 512-525). We declared that our study did not include longitudinal assessments, which could provide valuable insights into the temporal dynamics of placental changes and their impact on maternal and neonatal outcomes. Future prospective trials and mechanistic studies are needed to address this limitation.

Reviewer 3 Report

Comments and Suggestions for Authors

The work is of great practical and scientific interest, especially the study of a very common and insidious infection, which seemed to affect only the lungs, but turned out to have a much wider scale of damage, affecting many organs and systems.

The topic is very relevant and matches the content.

The purpose of the study is described unclearly and need to be changed into a more concrete form.

Please indicate the method you used to determine the sample size in this study.

Please indicate inclusion and exclusion criteria.

In statistical analysis, correct p<0.05 to p0.05.

Please record all results in M±SD format.

The main findings as related to the overall purpose of the study are discussed and explained in detail.

The conclusion should be short and specific, indicating the recommendation. Detailed explanation with Figure 8 needs to be moved to discussion

Author Response

The work is of great practical and scientific interest, especially the study of a very common and insidious infection, which seemed to affect only the lungs, but turned out to have a much wider scale of damage, affecting many organs and systems.

The topic is very relevant and matches the content.

The purpose of the study is described unclearly and need to be changed into a more concrete form.

Thanks for the reviewer’s suggestion. We clarified the aim of the study in more detail in the introduction section (Page 2, line 102-105).

Please indicate the method you used to determine the sample size in this study.

G*power was used to determine the sample size. This information was added in materials and methods - statistical analysis (Page 4, line 184-185).

Please indicate inclusion and exclusion criteria.

We included the inclusion and exclusion criteria in in materials and methods – study population (Page 3, line 136-146). Other comorbidities were stated also in this chapter.

In statistical analysis, correct p<0.05 to p≤0.05.

We thank the reviewer for this point and we have changed it in statistical analysis as recommended.

Please record all results in M±SD format.

We appreciate the reviewer's observation. Indeed, we presented the data in all figures using M±SD. However, for maternal age, gestational age, BMI, and Apgar score in the tables, we chose to report the median, interquartile range, and minimum-maximum values. This approach aligns with the common practice in medical research to describe pregnancy-related data.

The main findings as related to the overall purpose of the study are discussed and explained in detail.

We thank the reviewer for acknowledging our work.

The conclusion should be short and specific, indicating the recommendation. Detailed explanation with Figure 8 needs to be moved to discussion.

We appreciate and totally agree with the reviewer's feedback. We move the detailled description into the discussion part  (Page 14, line 482-505) because this is already an interpretation of the results. In addition, a more streamlined conclusion is added in the conclusion section (Page 16, line 528-532).

Reviewer 4 Report

Comments and Suggestions for Authors

The aim of this manuscript is to investigate the expression of inflammatory factors CCN1 and pNF-kB expression in the placenta of term pregnant women infected with SARS-CoV-2 compared to uninfected controls.

This manuscript provides a deep insight for some works: the study is within the journal’s scope, and I found it to be well-written, with rich contents. Even if the manuscript provides an organic overview, with a densely organized structure and based on well-synthetized evidence, there are some suggestions necessary to make the article complete and fully readable. For these reasons, the manuscript requires major changes.

Please find below an enumerated list of comments on my review of the manuscript:

MINOR POINTS:

The authors should provide a list of the abbreviations, mentioned in this manuscript.

MAJOR POINTS:

INTRODUCTION:

LINE 29: Coronavirus disease 2019 (COVID-19), declared a pandemic on 11 March 2020 by  the World Health Organization (WHO), showed as causative agent severe acute respiratory syndrome coronavirus-2 (SARS-CoV-2), an enveloped positive single-stranded RNA virus (see, for reference: Lu, R.; Zhao, X.; Li, J.; Niu, P.; Yang, B.; Wu, H.; Wang, W.; Song, H.; Huang, B.; Zhu, N.; et al. Genomic characterisation and epidemiology of 2019 novel coronavirus: Implications for virus origins and receptor binding. Lancet 2020395, 565–574). In this introductive section, the manuscript may benefit from providing a brief and complete background of the onset and progression of SARS-CoV-2 infection as a global health threat.

LINE 41: Histopathological data from placental tissues (fibrin deposition and inflammatory infiltrate in the intervillous space) are suggestive of a severe inflammatory response, with a consequent impairment of the fetal–maternal barrier. From an ultrastructural point of view, the virus targeted placental cells seem to be trophoblasts and fibroblasts. TEM data showed virus particles within the cytosol of placental cells. Furthermore, morphological data suggest that different factors (direct cytopathy, ischemic injuries an inflammatory response) cooperate in compromising the physiological functions of the placenta as gas exchange, metabolic transfer, hormone secretion, and fetal protection (see, for reference: Torge, D.; Bernardi, S.; Arcangeli, M.; Bianchi, S. Histopathological Features of SARS-CoV-2 in Extrapulmonary Organ Infection: A Systematic Review of Literature. Pathogens 202211, 867. https://doi.org/10.3390/pathogens11080867). This is the major concern of this manuscript: there is lack of structural and ultrastructural evidence regarding the effects of SARS-CoV-2 infection on the histopathology of the placenta villous tissues, as highlighted by recent data. This will improve the scientific impact of the manuscript.

The main topic is interesting, and certainly of great clinical impact. As regards the originality and strengths of this manuscript, this is a significant contribute to the ongoing research on this topic, as it extends the research field on the expression of inflammatory factors CCN1 and pNF-kB expression in the placenta of term pregnant women infected with SARS-CoV-2 compared to uninfected controls. Overall, the contents are rich, and the authors also give their deep insight for some works.

As regards the section of methods, there is a specific and detailed explanation for the methods used in this study: this is particularly significant, since the manuscript relies on a multitude of methodological and statistical analysis, to derive its conclusions. The methodology applied is overall correct, the results are reliable and adequately discussed.

The conclusion of this manuscript is perfectly in line with the main purpose of the paper: the authors have designed and conducted the study properly. As regards the conclusions, they are well written and present an adequate balance between the description of previous findings and the results presented by the authors.

Finally, this manuscript also shows a basic structure, properly divided and looks like very informative on this topic. Furthermore, figures and tables are complete, organized in an organic manner and easy to read.

In conclusion, this manuscript is densely presented and well organized, based on well-synthetized evidence. The authors were lucid in their style of writing, making it easy to read and understand the message, portrayed in the manuscript. Besides, the methodology design was appropriately implemented within the study. However, many of the topics are very concisely covered. This manuscript provided a comprehensive analysis of current knowledge in this field. Moreover, this research has futuristic importance and could be potential for future research. However, major concerns of this manuscript are with the introductive section: for these reasons, I have major comments for this section, for improvement before acceptance for publication. The article is accurate and provides relevant information on the topic and I have some major points to make, that may help to improve the quality of the current manuscript and maximize its scientific impact. I would accept this manuscript if the comments are addressed properly.

Author Response

The aim of this manuscript is to investigate the expression of inflammatory factors CCN1 and pNF-kB expression in the placenta of term pregnant women infected with SARS-CoV-2 compared to uninfected controls.

This manuscript provides a deep insight for some works: the study is within the journal’s scope, and I found it to be well-written, with rich contents. Even if the manuscript provides an organic overview, with a densely organized structure and based on well-synthetized evidence, there are some suggestions necessary to make the article complete and fully readable. For these reasons, the manuscript requires major changes.

Please find below an enumerated list of comments on my review of the manuscript:

MINOR POINTS:

The authors should provide a list of the abbreviations, mentioned in this manuscript.

We thank the reviewer for this point. The list of abbreviations was added (Page 16, line 535-544).

MAJOR POINTS:

INTRODUCTION:

LINE 29: Coronavirus disease 2019 (COVID-19), declared a pandemic on 11 March 2020 by  the World Health Organization (WHO), showed as causative agent severe acute respiratory syndrome coronavirus-2 (SARS-CoV-2), an enveloped positive single-stranded RNA virus (see, for reference: Lu, R.; Zhao, X.; Li, J.; Niu, P.; Yang, B.; Wu, H.; Wang, W.; Song, H.; Huang, B.; Zhu, N.; et al. Genomic characterisation and epidemiology of 2019 novel coronavirus: Implications for virus origins and receptor binding. Lancet 2020, 395, 565–574). In this introductive section, the manuscript may benefit from providing a brief and complete background of the onset and progression of SARS-CoV-2 infection as a global health threat.

We thank the reviewer for the valuable comment and we added this information in the introduction section (Page 1, line 29-32).

LINE 41: Histopathological data from placental tissues (fibrin deposition and inflammatory infiltrate in the intervillous space) are suggestive of a severe inflammatory response, with a consequent impairment of the fetal–maternal barrier. From an ultrastructural point of view, the virus targeted placental cells seem to be trophoblasts and fibroblasts. TEM data showed virus particles within the cytosol of placental cells. Furthermore, morphological data suggest that different factors (direct cytopathy, ischemic injuries an inflammatory response) cooperate in compromising the physiological functions of the placenta as gas exchange, metabolic transfer, hormone secretion, and fetal protection (see, for reference: Torge, D.; Bernardi, S.; Arcangeli, M.; Bianchi, S. Histopathological Features of SARS-CoV-2 in Extrapulmonary Organ Infection: A Systematic Review of Literature. Pathogens 2022, 11, 867. https://doi.org/10.3390/pathogens11080867). This is the major concern of this manuscript: there is lack of structural and ultrastructural evidence regarding the effects of SARS-CoV-2 infection on the histopathology of the placenta villous tissues, as highlighted by recent data. This will improve the scientific impact of the manuscript.

We gratefully acknowledge the reviewer's suggestion and completely agree that further ultrastructural evidence of SARS-Cov2 infection would confirm its effects on placental tissue. This information was added in the introduction section (Page 2, line 56-64).

The main topic is interesting, and certainly of great clinical impact. As regards the originality and strengths of this manuscript, this is a significant contribute to the ongoing research on this topic, as it extends the research field on the expression of inflammatory factors CCN1 and pNF-kB expression in the placenta of term pregnant women infected with SARS-CoV-2 compared to uninfected controls. Overall, the contents are rich, and the authors also give their deep insight for some works.

As regards the section of methods, there is a specific and detailed explanation for the methods used in this study: this is particularly significant, since the manuscript relies on a multitude of methodological and statistical analysis, to derive its conclusions. The methodology applied is overall correct, the results are reliable and adequately discussed.

The conclusion of this manuscript is perfectly in line with the main purpose of the paper: the authors have designed and conducted the study properly. As regards the conclusions, they are well written and present an adequate balance between the description of previous findings and the results presented by the authors.

Finally, this manuscript also shows a basic structure, properly divided and looks like very informative on this topic. Furthermore, figures and tables are complete, organized in an organic manner and easy to read.

In conclusion, this manuscript is densely presented and well organized, based on well-synthetized evidence. The authors were lucid in their style of writing, making it easy to read and understand the message, portrayed in the manuscript. Besides, the methodology design was appropriately implemented within the study. However, many of the topics are very concisely covered. This manuscript provided a comprehensive analysis of current knowledge in this field. Moreover, this research has futuristic importance and could be potential for future research. However, major concerns of this manuscript are with the introductive section: for these reasons, I have major comments for this section, for improvement before acceptance for publication. The article is accurate and provides relevant information on the topic and I have some major points to make, that may help to improve the quality of the current manuscript and maximize its scientific impact. I would accept this manuscript if the comments are addressed properly.

We thank the reviewer for acknowledging our work and the kind comments.

Round 2

Reviewer 2 Report

Comments and Suggestions for Authors

The authors commplied with the recommandation. The article can be published in the current form

Comments on the Quality of English Language

English is OK

Reviewer 3 Report

Comments and Suggestions for Authors

Indeed, the manuscript has become more interesting and informative for readers.

High relevance of the topic. The title matches the content. Materials and methods are described in detail. The discussion and conclusion were based on the results obtained.

Reviewer 4 Report

Comments and Suggestions for Authors

The authors have significantly improved the manuscript.